# Coordination of Cyclic Electron Flow and Water–Water Cycle Facilitates Photoprotection under Fluctuating Light and Temperature Stress in the Epiphytic Orchid *Dendrobium officinale*

**DOI:** 10.3390/plants10030606

**Published:** 2021-03-23

**Authors:** Hu Sun, Qi Shi, Shi-Bao Zhang, Wei Huang

**Affiliations:** 1Kunming Institute of Botany, Chinese Academy of Sciences, Kunming 650201, China; sunhu19@mails.ucas.ac.cn (H.S.); shiqi17@email.swu.edu.cn (Q.S.); sbzhang@mail.kib.ac.cn (S.-B.Z.); 2University of Chinese Academy of Sciences, Beijing 100049, China; 3College of Horticulture and Landscape Architecture, Southwest University, Chongqing 400715, China; 4Bio-Innovation Center of DR PLANT, Kunming Institute of Botany, Chinese Academy of Sciences, Kunming 650201, China

**Keywords:** *Dendrobium officinale*, cyclic electron flow, water–water cycle, photosynthesis, photoprotection

## Abstract

Photosystem I (PSI) is the primary target of photoinhibition under fluctuating light (FL). Photosynthetic organisms employ alternative electron flows to protect PSI under FL. However, the understanding of the coordination of alternative electron flows under FL at temperature stresses is limited. To address this question, we measured the chlorophyll fluorescence, P700 redox state, and electrochromic shift signal in leaves of *Dendrobium officinale* exposed to FL at 42 °C, 25 °C, and 4 °C. Upon a sudden increase in illumination at 42 °C and 25 °C, the water–water cycle (WWC) consumed a significant fraction of the extra reducing power, and thus avoided an over-reduction of PSI. However, WWC was inactivated at 4 °C, leading to an over-reduction of PSI within the first seconds after light increased. Therefore, the role of WWC under FL is largely dependent on temperature conditions. After an abrupt increase in light intensity, cyclic electron flow (CEF) around PSI was stimulated at any temperature. Therefore, CEF and WWC showed different temperature responses under FL. Furthermore, the enhancement of CEF and WWC at 42 °C quickly generated a sufficient trans-thylakoid proton gradient (ΔpH). The inactivation of WWC at 4 °C was partially compensated for by an increased CEF activity. These findings indicate that CEF and WWC coordinate to protect PSI under FL at temperature stresses.

## 1. Introduction

Photosynthetic organisms usually experience dynamic fluctuations of light intensity under natural field conditions [1]. During fluctuating light (FL), a sudden increase in light intensity leads to a rapid increase in electron flow from photosystem II (PSII) to PSI [2]. However, the excited states in PSI cannot be immediately consumed by the dark reactions because CO_2_ fixation has a much slower kinetics [3], generating the imbalance between electron supply and nicotinamide adenine dinucleotide phosphate (NADPH) consumption [4]. The resulting excess excitation energy induces the production of reactive oxygen species (ROS) within PSI, increasing the risk of PSI photoinhibition [5,6,7]. Once PSI is damaged, the CO_2_ assimilation rate significantly decreases, impairing plant growth [8,9,10,11,12,13,14]. Fortunately, plants have evolved several effective mechanisms to protect PSI against photo-oxidative damage under FL [15,16,17]. In angiosperms, two alternative electron flows, cyclic electron flow (CEF) around PSI and water–water cycle (WWC), have the potential to fine-tune the redox state of PSI under FL [6,18,19,20].

During CEF, ΔpH is generated without production of NAPDH [21,22]. The CEF-dependent ΔpH generation can control the electron flow to PSI and thus protect PSI at donor side [23,24,25,26]. Furthermore, an increased ΔpH can provide additional ATP and thus increase the ATP/NADPH production ratio [27,28]. This adjustment of the ATP/NADPH production ratio favors the operation of CO_2_ assimilation and photorespiration, sustaining electron sink downstream of PSI and alleviating an over-reduction of PSI electron carriers [6]. During WWC, electrons derived from water are transported through photosynthetic electron chain to O_2_, and the resulting O_2_^-^ is scavenged by superoxide dismutase and ascorbate peroxidase [29]. The operation of WWC can consume the excess exited electrons in PSI and thus avoids an over-reduction of PSI [18]. Although WWC is more effective in protecting PSI under FL than CEF, the activity of WWC in angiosperms is species dependent [19]. In model plants *Arabidopsis thaliana* and *Nicotiana tabacum*, the role of WWC in regulation of photosynthetic electron flow under FL was negligible [5,30,31]. By comparison, WWC significantly regulated the redox state of PSI under FL in *Camellia* species [19], *Bryophyllum pinnatum* [32], and *Dendrobium officinale* [33]. However, the coordination of CEF and WWC under FL at temperature stresses are poorly understood.

Upon transition from low to high light, CEF activity first stimulated and then gradually decreased [5,34]. Although CEF favored the rapid formation of ΔpH, such transient stimulation could not prevent a transient over-reduction of PSI in *Arabidopsis thaliana* and tobacco because they could not generate an enough ΔpH to fully activate photosynthetic control at cytochrome b6f [19,35]. Such transient over-reduction of PSI led to significant photoinhibition of PSI under FL for leaves of *A. thaliana* and tobacco. During prolonged exposure to high light, ΔpH gradually increased to the optimum value, optimizing the redox state of PSI and thus preventing uncontrolled photoinhibition of PSI [35]. At 25 °C and 42 °C, *D. officinale* also showed a transient stimulation of CEF upon an abrupt increase in light intensity [33,36]. Meanwhile, *D. officinale* displayed a rapid re-oxidation of P700. Such rapid re-oxidation of P700 was not observed when measured under anaerobic conditions [33]. Therefore, the operation of WWC is responsible for the rapid oxidation of PSI under FL. As we know, CEF-dependent ΔpH formation protects PSI under FL at donor and acceptor side. However, it is unclear whether the synchronous stimulation of CEF and WWC can form an enough ΔpH to protect PSI at donor side.

Interestingly, upon dark-to-light transition, the rapid re-oxidation of P700 in *D. officinale* was clearly missing at the chilling temperature of 4 °C, indicating the inactivation of WWC at low temperature [37]. Under such conditions, the severe over-reduction of PSI should be rescued by the formation of ΔpH. Concomitantly, linear electron flow was also highly restricted and therefore its role in ΔpH formation was small [37]. Furthermore, an over-reduction of PSI electron carriers can trigger the stimulation of CEF under FL [38]. Therefore, we speculated that, upon any increase in light intensity at low temperature, *D. officinale* might display a highly stimulation of CEF to compensate for the inactivation of WWC.

In this study, we measured the chlorophyll fluorescence, P700 redox state, and electrochromic shift signal in leaves of *Dendrobium officinale* exposed to FL at 42 °C, 25 °C, and 4 °C. *D. officinale* is native to subtropical forests in China (elevation < 1600 m) and grows on semi-humid rocks. Under natural field conditions, *D. officinale* usually experiences FL and high temperature in summer but undergoes FL and chilling temperature in winter. Our aims were to (1) evaluate the temperature responses of CEF and WWC under FL, and (2) assess whether CEF and WWC coordinate to fine-tune the redox state of PSI under FL at temperature stresses.

## 2. Results

### 2.1. Photosynthetic Regulation after Transition from Dark to Light

We first examined the changes in redox kinetics of the PSI primary electron donor (P700) upon transition from dark to actinic light (AL; 1,455 μmol photons m^−2^ s^−1^) at 42 °C, 25 °C, and 4 °C (Figure 1). At 25 °C and 42 °C, *D. officinale* showed a rapid re-oxidation of P700 in 5 s and 20 s, respectively. By comparison, such rapid re-oxidation was not observed at 4 °C. Because the rapid re-oxidation of P700 upon dark-to-light transition in *D. officinale* was caused by the operation of WWC [33], these results indicated that WWC activity was enhanced at 42 °C but was inactivated at 4 °C.

After transition from dark to AL for 10 s and 30 s, the values of quantum yield of PSI photochemistry [Y(I)] were higher at 42 °C when compared with 25 °C and 4 °C (Figure 2A). After light acclimation for 11 min, the value of Y(I) at 25 °C was much higher than that at 4 °C and 42 °C. After this light transition for 10 s, *D. officinale* displayed high values of Y(ND) (PSI donor side limitation) at 25 °C and 42 °C (Figure 2B), suggesting that PSI was highly oxidized. However, such rapid oxidation of PSI was not observed when measured at 4 °C (Figure 2B). Within the first 2 min after transition from dark to AL at 4 °C, Y(ND) gradually increased from 0.03 to 0.54 (Figure 2B). Meanwhile, the value of Y(NA) (PSI acceptor side limitation) gradually decreased from 0.9 to 0.37 (Figure 2C), indicating the over-reduction of PSI in *D. officinale* upon dark-to-light transition at 4 °C. By comparison, such over-reduction of PSI was not observed when measured at 25 °C and 42 °C, as indicated by the low values of Y(NA) (Figure 2C).

After transition from dark to AL for 10 s and 30 s, the effective quantum yield of PSII [Y(II)] at 42 °C was much higher than those values at 25 °C and 4 °C (Figure 2D). Subsequently, Y(II) gradually decreased at 42 °C but gradually increased at 25 °C and 4 °C. The induction rate of non-photochemical quenching (NPQ) was positive to measurement temperature (Figure 2E). However, after photosynthetic induction for 11 min, NPQ capacity was significantly inhibited at 4 °C and 42 °C compared with at 25 °C. Owing to the relatively lower NPQ at 4 °C and 42 °C, the quantum yield of non-regulated energy dissipation in PSII [Y(NO)] was higher at 4 °C and 42 °C than at 25 °C (Figure 2F).

Within the first 30 s after transition from dark to AL, values for the photosynthetic transport rate through PSI (ETRI) at 42 °C were significantly higher than those at 25 °C and 4 °C (Figure 3A), owing to the higher ETRII value at 42 °C (Figure 3B). Subsequently, ETRI was significantly inhibited at either 42 °C or 4 °C (Figure 3A), mainly due to the restriction of ETRII (Figure 3B). The performance of CEF after transition from dark to AL significantly differed at different temperature conditions. At 42 °C, CEF was highly activated within 10 s and then gradually decreased to the minimal level (Figure 3C). At 4 °C, CEF gradually increased to the maximal level in 60 s and then gradually decreased to the steady state (Figure 3C). The CEF activation at 4 °C was higher than that at 42 °C.

Within the first 10 s after transition from dark to AL, *D. officinale* could not generate a sufficient trans-thylakoid proton gradient (ΔpH) at 25 °C and 4 °C (Figure 4A). Meanwhile, a high level of ΔpH was generated at 42 °C (Figure 4A). During photosynthetic induction, chloroplast ATP activity (*g*_H_^+^) gradually increased at 25 °C but gradually decreased at 42 °C (Figure 4A). At 4 °C, the low level of *g*_H_^+^ was maintained as stable (Figure 4A). Therefore, the kinetics of ΔpH and *g*_H_^+^ upon dark-to-light transition were largely affected by temperature conditions.

### 2.2. Photosynthetic Regulation after Transition from Low to High Light

At a low light of 59 μmol photons m^−2^ s^−1^, *D. officinale* showed higher Y(I), lower Y(ND), and higher Y(NA) at 25 °C than those at 4 °C and 42 °C (Figure 5). After transition from 59 to 1455 μmol photons m^−2^ s^−1^, Y(I) rapidly decreased (Figure 5A), and Y(ND) rapidly increased. Meanwhile, Y(NA) rapidly decreased at 25 °C and 42 °C (Figure 5C). However, Y(NA) first increased and then decreased to the steady state at 4 °C (Figure 5C). Therefore, the electron sink downstream of PSI was significantly inhibited in *D. officinale* when exposed to the low temperature of 4 °C. The chlorophyll fluorescence measurement indicated that Y(II) under high light was inhibited at both 4 °C and 42 °C (Figure 5D). At low light, NPQ was enhanced at 4 °C compared with at 25 °C and 42 °C (Figure 5E). After transition from low to high light, NPQ rapidly increased at 25 °C and 42 °C but changed little at 4 °C (Figure 5E). Under high light, the inhibition of NPQ induction at 4 °C and 42 °C led to the increase in Y(NO) (Figure 5F), increasing the production of ROS within PSII and accelerating PSII photoinhibition.

After this light transition, ETRI increased largely at 25 °C but slightly at 4 °C and 42 °C (Figure 6A). By comparison, ETRII significantly increased at 25 °C and 42 °C but changed little at 4 °C (Figure 6B). The value of ETRI–ETRII rapidly increased and then gradually decreased at any temperature, indicating the stimulation of CEF within the first seconds after light increased (Figure 6C). Furthermore, such stimulation of CEF was stronger at 25 °C and 4 °C compared with at 42 °C (Figure 6C). Similar to ETRI–ETRII, the ETRI/ETRII ratio first increased and then gradually decreased (Figure 7). Moreover, the values of ETRI/ETRII ratio at 4 °C were much higher than those at 25 °C and 42 °C (Figure 7), indicating that the contribution of CEF to total photosynthetic electron transport was enhanced under FL at 4 °C.

## 3. Discussion

Under natural field conditions, FL is a common light condition experienced by plants. Upon a sudden increase in light intensity, the rapid increase in electron flow to PSI could not be immediately consumed by the Calvin cycle [4]. The resulting excess excited electrons accumulated in PSI, leading to the production of ROS within PSI and thus increasing the risk of PSI photoinhibition [6,39,40]. In order to protect PSI under FL, plants employ alternative electron flows to regulate the redox state of PSI. In organisms from cyanobacteria up to gymnosperms, flavodiiron proteins (FDPs) rapidly accept electrons from PSI to O_2_, resulting in the fast oxidation of PSI [4,41,42,43,44]. Because oxidized P700 (P700^+^) is able to induce non-photochemical energy dissipation in PSI, FDPs act as safety valve for photosynthetic apparatus. However, angiosperms lost FDPs during evolution and they alternatively used CEF to fine-tune photosynthetic electron transport chain under FL [39]. In addition, some angiosperms have high activities of WWC, which alleviates an over-reduction of PSI and thus protects PSI under FL [19,32,33]. However, the understanding of the coordination of CEF and WWC under FL at temperature stresses is limited.

In this article, we demonstrated that the angiosperm *D. officinale* significantly showed a transient stimulation of CEF for a few seconds after an increase in illumination at 42 °C, 25 °C, and 4 °C (Figure 3C). Such transient stimulation of CEF favored the rapid formation of ΔpH, strengthening the photosynthetic control at cytochrome b6f and enhancing the ATP/NADPH production ratio. At 25 °C and 42 °C, *D. officinale* showed a rapid re-oxidation of P700 upon transition from dark to AL (Figure 1), indicating the rapid electron sink downstream of PSI. It has been indicated that WWC is responsible for the rapid P700 re-oxidation in *D. officinale* [33]. Furthermore, the re-oxidation of P700 at 42 °C was faster than that at 25 °C, indicating that the activity of WWC was enhanced at 42 °C. By comparison, a rapid re-oxidation of P700 upon transition from dark to AL was clearly missing at 4 °C (Figure 1), suggesting the inactivation of WWC at 4 °C. Therefore, the electron transfer from PSI to O_2_ in WWC is a typical temperature-dependent reaction.

Low temperature largely restricts the light use efficiency, leading to an increase in excess light energy. As a result, chilling-light stress can cause photoinhibition of PSII and/or PSI in chilling-sensitive plants [45,46,47,48,49]. However, the photosynthetic regulation under FL at low temperature is little known. Upon a sudden transition from dark to AL at 4 °C, the inactivation of WWC activity and an insufficient ΔpH formation led to severe over-reduction of PSI, although ETRII was largely inhibited (Figure 2C and Figure 3B). Under such conditions, CEF was highly activated and CEF presented the major fraction of the total electron flow (Figure 3C). Therefore, CEF largely contributed to the formation of ΔpH upon dark-to-light transition at 4 °C. After exposure to AL for 4 min at 4 °C, *D. officinale* generated an enough ΔpH (Figure 4A), which optimized the redox state of PSI (Figure 2C). Upon transition from low to high light, the inactivation of WWC at 4 °C led to a transient increase of PSI acceptor side limitation (Figure 5C). As we know, PSI induced by FL is mainly caused by the over-reduction of PSI electron carriers. Therefore, FL at 4 °C can cause a significant photoinhibiton of PSI in *D. officinale*. Under such conditions, the CEF activity was highly stimulated (Figure 6C), and the contribution of CEF to total electron transport rate was enhanced (Figure 7). Therefore, the inactivation of WWC at 4 °C under FL was highly compensated for by the stimulation of CEF, and CEF played a major role in adjustment of PSI redox state under FL at 4 °C. It has been documented that there are two major pathways responsible for CEF: PGR5/PGRL1 and NDH. At normal growth temperature, either PGR5/PGRL1 or NDH pathway plays a significant role in photoprotection under FL [6,20,39,40]; their responses and roles under FL at low temperatures need to be clarified in future study.

In contrast to the phenotype at 4 °C, *D. officinale* showed a rapid oxidation of PSI after any increase in light intensity at 25 °C. After transition from dark to AL for 10 s at 25 °C, *D. officinale* could not build up an enough ΔpH (Figure 4A). Thus, the rapid oxidation of PSI under FL at 25 °C could not be explained by the formation of ΔpH but was caused by the electron sink downstream of PSI via WWC. During WWC, electrons transported from PSII to PSI can be consumed by photo-reduction of O_2_, making PSI oxidized and avoiding an over-reduction of PSI electron carriers. Therefore, WWC regulated the redox state of PSI at acceptor side more than donor side at 25 °C. Because PSI becomes damaged only when PSI electron carriers are over-reduced under high light [6,38], the WWC-dependent oxidation of PSI prevented PSI photoinhibition under FL at 25 °C. Meanwhile, the stimulation of CEF was also observed, which helped the formation of ΔpH along with WWC. A rapid formation of ΔpH adjusted the ATP/NADPH production ratio and thus favored the Calvin cycle and photorespiration. Therefore, when exposed to FL at 25 °C, WWC rapidly regulated PSI redox state and CEF adjusted energy balance to sustain carbon fixation.

Moderate heat stress is a typical climatic condition in summer. Under moderate heat stress, the oxygen-evolving complex encounters photodamage and the carbon fixation is suppressed, leading to severe photoinhibition of PSII [50,51,52,53,54]. During drought and heat wave stress, alternative electron sinks were enhanced in European beech to utilize the excess light energy and decrease the excitation pressure on PSII [55]. Furthermore, moderate heat stress increased the thylakoid proton conductance and thus decreased the ΔpH formation [31,56,57]. The resulting over-reduction of PSI electron carriers promotes the production of ROS within PSI and thus causes PSI photoinhibition [11,58]. However, photosynthetic regulation under FL at moderate heat stress has not yet been clarified. Comparing with the phenotype at 25 °C, the activity of WWC was significantly enhanced at 42 °C. Meanwhile, *D. officinale* showed a high level of CEF (Figure 3C). The synchronous enhancement of WWC and CEF led to the rapid formation of a high ΔpH at 42 °C (Figure 4A). Therefore, upon dark-to-light transition at 42 °C, WWC optimized PSI redox state at donor and acceptor sides. Upon transition from low to high light, *D. officinale* showed much higher ETRII at 42 °C than at 4 °C (Figure 6B). Meanwhile, PSI acceptor side limitation decreased at 42 °C but increased at 4 °C (Figure 5C). These results indicated that the difference in PSI redox kinetics under FL between 42 °C and 4 °C was not caused by the change in ETRII but was determined by the temperature responses of WWC. Although CEF was highly stimulated at 4 °C, the operation of WWC was more effective in oxidizing PSI under FL.

## 4. Materials and Methods

### 4.1. Plant Materials

Our previous studies indicated that the epiphytic orchid *Dendrobium officinale* showed a significant electron flow via WWC [33,36]. In order to examine the coordination of WWC and CEF under FL, we chose *D. officinale* for experiments in the present study. Plants grew in a greenhouse with moderate relative air humidity (60–70%) without water or nutrition stress. During the period of measurement, the growth temperature was 25/10 °C at day/night. We used non-woven shade to control the light condition being 40% of full sunlight, and the maximum photosynthetic photons flux density at daytime was approximately 800 μmol photons m^−2^ s^−1^. The fully expanded but not senescence leaves were used for measurements.

### 4.2. P700 Redox Kinetics Measurement

To evaluate the alternative electron flow rather than CO_2_ fixation, we dark adapted attached leaves for 60 min to fully inactivate the Calvin cycle. Afterward, we used a Dual-PAM 100 measuring system (Heinz Walz, Effeltrich, Germany) to record the changes in P700 redox kinetics after transition from dark to 1455 μmol photons m^−2^ s^−1^ for 20 s at 42 °C, 25 °C, and 4 °C.

### 4.3. PSI and PSII Measurements

We used a Dual-PAM 100 measuring system (Heinz Walz) to synchronously record the PSI and PSII parameters [59]. Before measurements, plants were incubated at 42 °C, 25 °C, and 4 °C for 15 min in the dark. Afterwards, photosynthetic measurements were conducted at each temperature. In the present study, actinic light from a 635 nm light-emitting diode (LED) equipped in Dual-PAM 100 was used as actinic light for photosynthetic measurements. The PSI photosynthetic parameters were measured on the basis of P700 oxidation signal (difference of intensities of 830 and 875 nm pulse-modulated measuring light reaching the photodetector) [59]. The P700^+^ signals (*P*) may vary between a minimal (P700 fully reduced) and a maximal level (P700 fully oxidized). The maximum level (*P*_m_) was determined with application of a saturation pulse (300 ms and 10,000 μmol photons m^−2^ s^−1^) after pre-illumination with far-red light. *P*_m_*’* was determined similar to *P*_m_ but with actinic light instead of far-red light. The PSI parameters were calculated as follows: the quantum yield of PSI photochemistry, Y(I) = (*P*_m_*′* − *P*)/*P*_m_; the quantum yield of PSI non-photochemical energy dissipation due to the donor side limitation, Y(ND) = *P*/*P*_m_; the quantum yield of the non-photochemical energy dissipation due to the acceptor side limitation, Y(NA) = (*P*_m_ − *P*_m_*′*)/*P*_m_.

The PSII parameters were calculated as follows [60,61]: the effective quantum yield of PSII photochemistry, Y(II) = (*F_m_′* − *F_s_*)/*F_m_′*; non-photochemical quenching in PSII, NPQ = (*F_m_* − *F_m_′*)/*F_m_′*; the quantum yield of non-regulated energy dissipation in PSII, Y(NO) = *F_s_*/*F_m_*. *F_m_* and *F_m_’* are the maximum fluorescence intensity after dark or light adaptation, respectively. *F_s_* is the steady state fluorescence intensity after light adaptation. *F_m_* and *F_m_′* were determined using a saturation pulse (300 ms and 10,000 μmol photons m^−2^ s^−1^, respectively). The electron transport rates through PSI and PSII were calculated as ETRI (or ETRII) = PPFD × Y(I) (or Y(II)) × 0.84 × 0.5, where PPFD is photosynthetic photon flux density.

### 4.4. Proton Motive Force Measurement

We used a dual PAM-100 (Heinz Walz) equipped with a P515/535 emitter-detector module to monitor the electrochromic shift (ECS) signal as the change in absorbance at 515 nm [62,63,64]. After the dark adaptation for 60 min, ECS_ST_ was measured by a single turnover flash. Subsequently, leaves were illuminated at 1455 μmol photons m^−2^ s^−1^ and ΔpH were measured. The ECS dark interval relaxation kinetics (DIRK_ECS_) were used to calculate the ΔpH values [65], in which all ΔpH levels were normalized against the magnitude of ECS_ST_.

### 4.5. FL Treatment

After photosynthetic induction at 1455 μmol photons m^−2^ s^−1^ for 11 min to activate photosynthetic electron sinks, leaves were exposed to FL alternating between 59 and 1455 μmol photons m^−2^ s^−1^ every 2 min.

### 4.6. Statistical Analysis

All results are shown as mean values of 5 leaves from 5 individual plants. An independent *t*-test was to determine the significant differences in PSI and PSII parameters between different temperatures (*α* = 0.05). One-way ANOVA and a post hoc test were used to determine the differences in ΔpH and *g*_H_^+^ between different treatments (*α* = 0.05).

## 5. Conclusions

We found that the major alternative electron flows, CEF and WWC, showed different temperature responses under FL. Specifically, CEF is a universal but WWC is a temperature-specific protective mechanism under FL at temperature stresses. This finding may explain why WWC is not a common alternative electron flow used by angiosperms to cope with FL under natural field conditions. Furthermore, the highly stimulation of CEF compensated for the inactivation of WWC under FL at 4 °C. Taken together, the coordination of CEF and WWC protected PSI photoinhibition under FL at temperature stresses.

## Figures and Tables

**Figure 1 plants-10-00606-f001:**
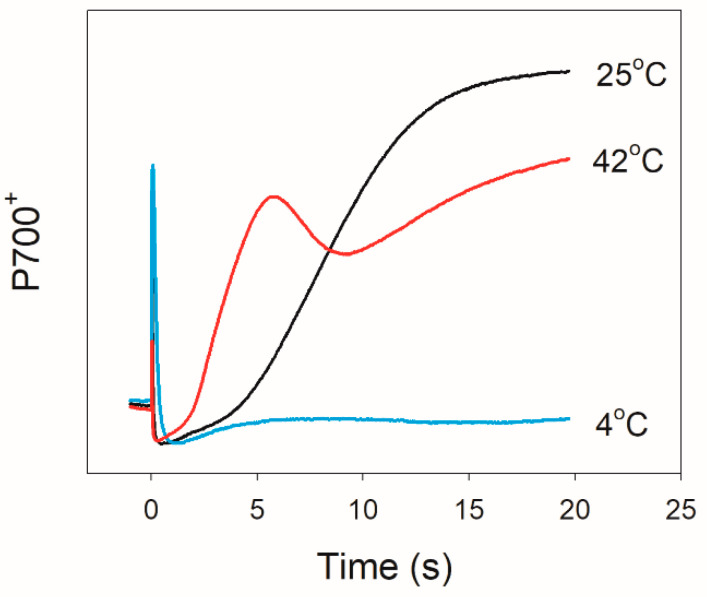
Changes in P700 redox kinetics after transition from dark to actinic light (1455 μmol photons m^−2^ s^−1^) at 25 °C, 4 °C, and 42 °C. Data are mean values of five leaves from five individual plants.

**Figure 2 plants-10-00606-f002:**
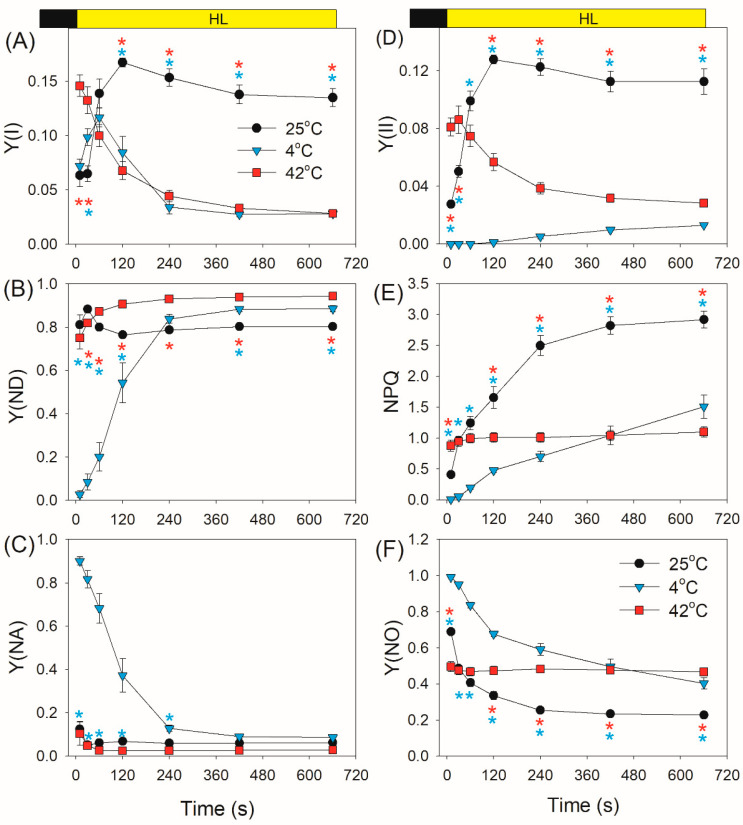
Changes in photosystem (PS) I and PSII parameters after transition from dark to actinic light (1455 μmol photons m^−2^ s^−1^) at 25 °C, 4 °C, and 42 °C. (**A**–**C**) Y(I) represents the quantum yield of PSI photochemistry; Y(ND), the quantum yield of PSI non-photochemical energy dissipation due to the donor side limitation; Y(NA), the quantum yield of the non-photochemical energy dissipation due to the acceptor side limitation. (**D**–**F**) Y(II) represents the effective quantum yield of PSII photochemistry; NPQ, non-photochemical quenching in PSII; Y(NO), the quantum yield of non-regulated energy dissipation in PSII. Values are means ± standard error (SE) (*n* = 5). Red asterisks indicate significant differences between 25 °C and 42 °C, and blue asterisks indicate significant differences between 25 °C and 4 °C.

**Figure 3 plants-10-00606-f003:**
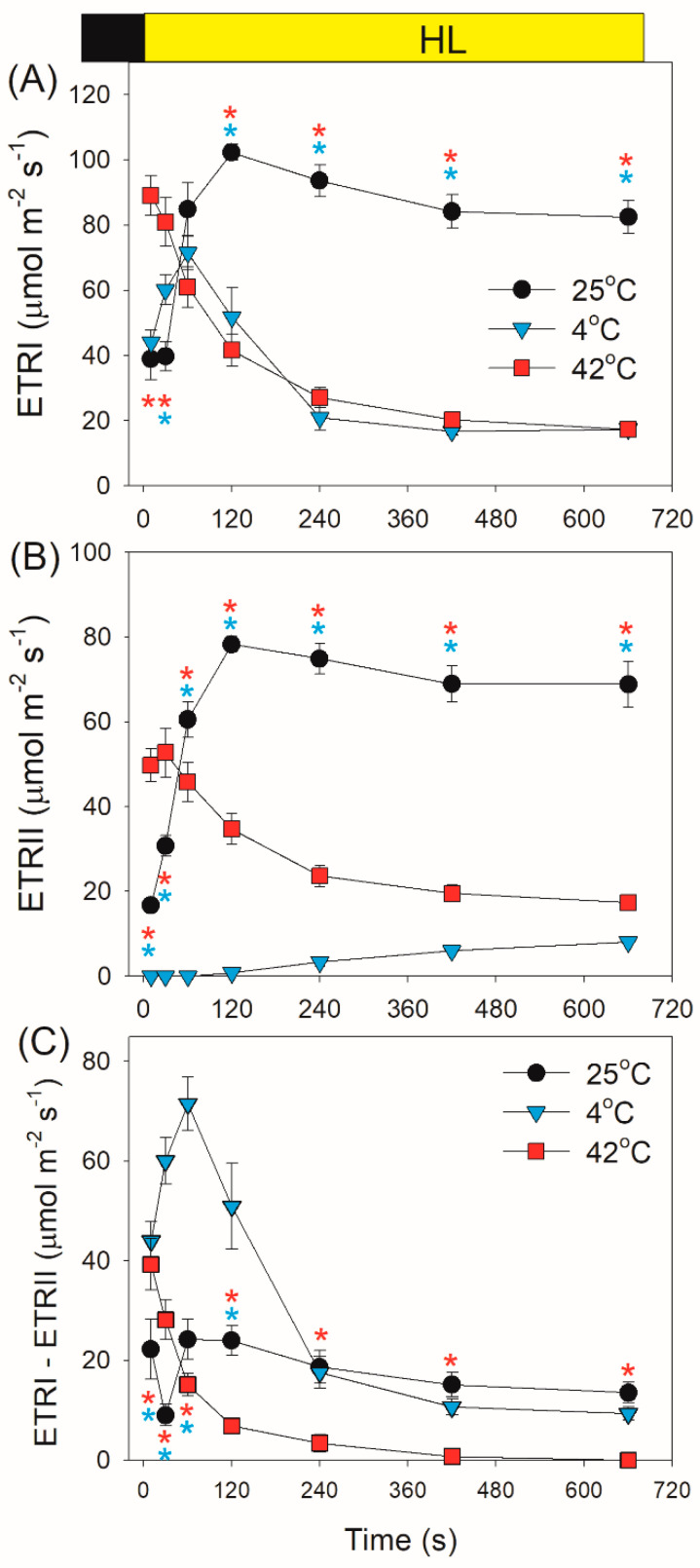
Changes in photosynthetic electron transport rates after transition from dark to actinic light (1455 μmol photons m^−2^ s^−1^) at 25 °C, 4 °C, and 42 °C. (**A**–**C**) ETRI represents the photosynthetic transport rate through PSI; ETRII, the photosynthetic transport rate through PSII; ETRI–ETRII, the rate of cyclic electron flow (CEF). Values are means ± SE (*n* = 5). Red asterisks indicate significant differences between 25 °C and 42 °C, and blue asterisks indicate significant differences between 25 °C and 4 °C.

**Figure 4 plants-10-00606-f004:**
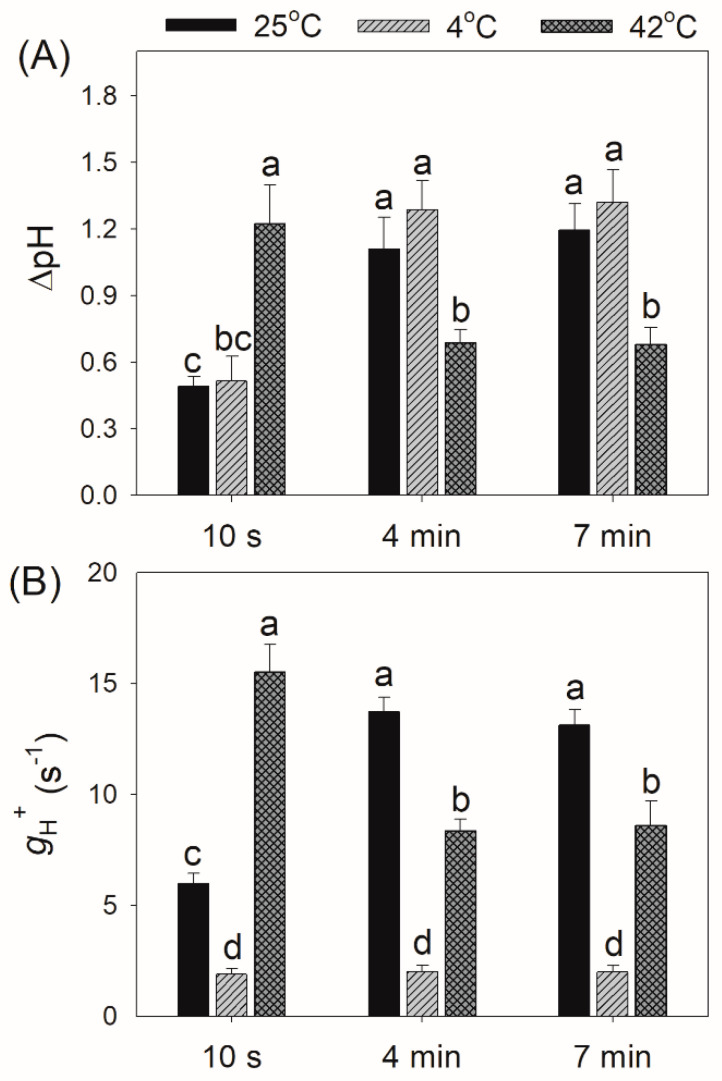
Changes in trans-thylakoid proton gradient (ΔpH) (**A**) and chloroplast ATP synthase activity (*g*_H_^+^) (**B**) after transition from dark to actinic light (1455 μmol photons m^−2^ s^−1^) at 25 °C, 4 °C, and 42 °C. Values are means ± SE (*n* = 5). Different letters indicate significant differences among different treatments.

**Figure 5 plants-10-00606-f005:**
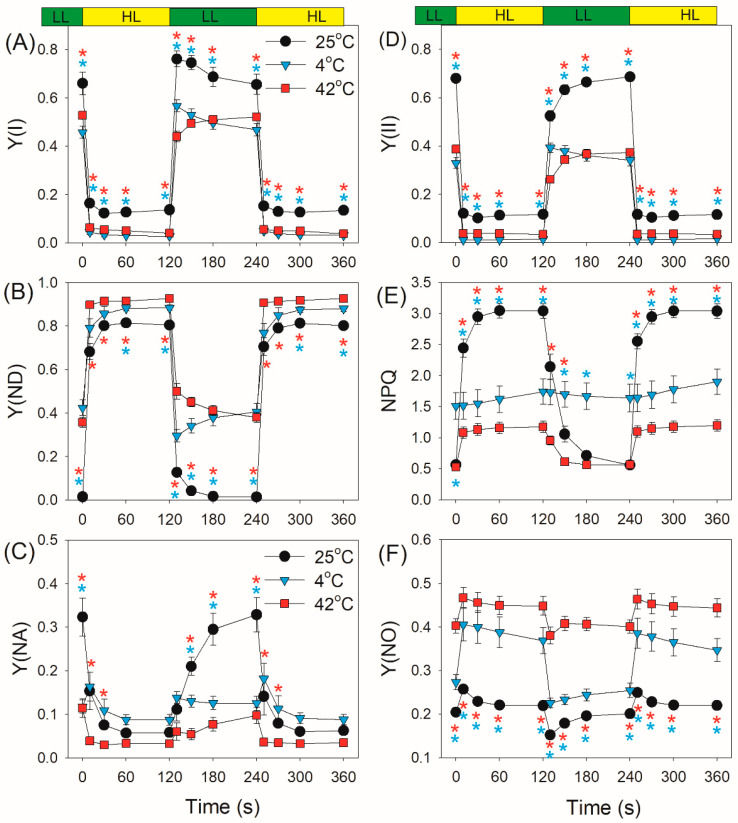
Changes in PSI and PSII parameters after transition from 59 to 1455 μmol photons m^−2^ s^−1^ at 25 °C, 4 °C, and 42 °C. (**A**–**C**) Y(I), Y(ND) and Y(NA); (**D**–**F**) Y(II), NPQ and Y(NO). Values are means ± SE (*n* = 5). Red asterisks indicate significant differences between 25 °C and 42 °C, and blue asterisks indicate significant differences between 25 °C and 4 °C.

**Figure 6 plants-10-00606-f006:**
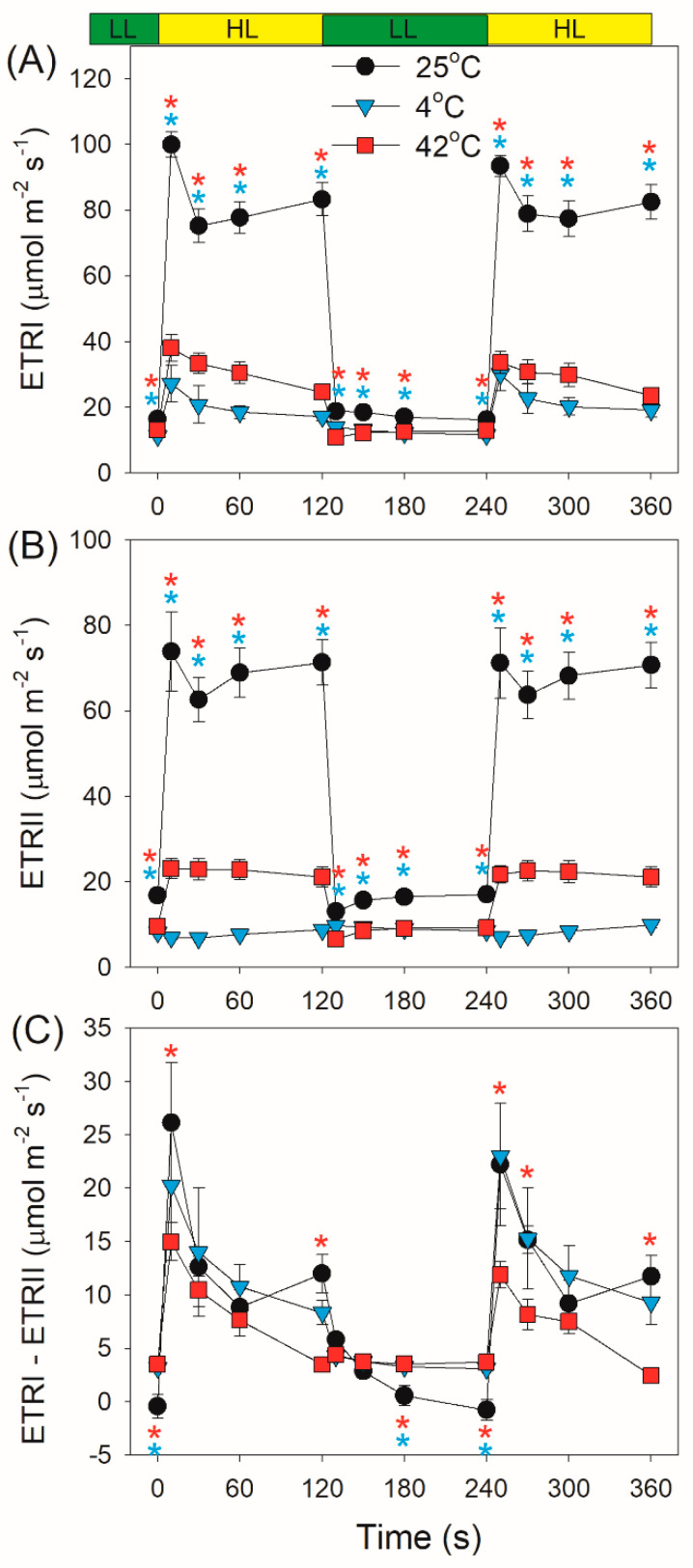
Changes in ETRI (**A**), ETRII (**B**) and ETRI–RTRII (**C**) after transition from 59 to 1455 μmol photons m^−2^ s^−1^ at 25 °C, 4 °C, and 42 °C. Values are means ± SE (*n* = 5). Red asterisks indicate significant differences between 25 °C and 42 °C, and blue asterisks indicate significant differences between 25 °C and 4 °C.

**Figure 7 plants-10-00606-f007:**
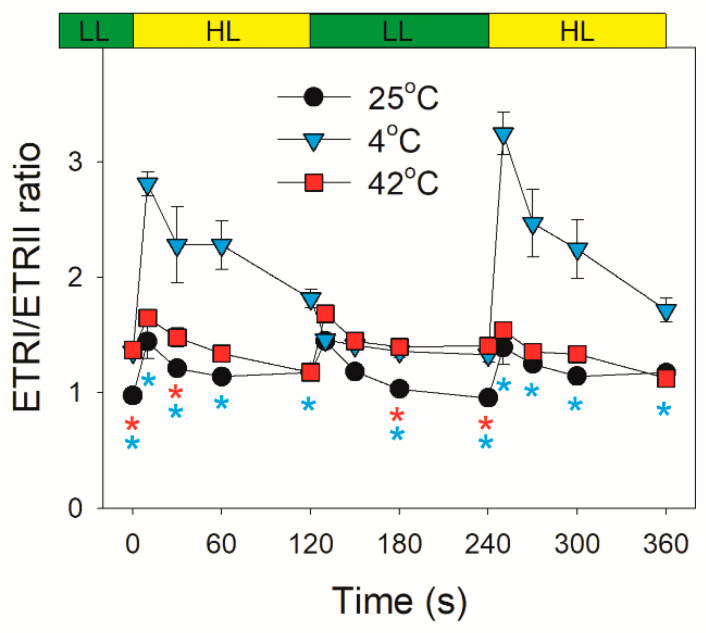
Changes in the ETRI/ETRII ratio after transition from 59 to 1455 μmol photons m^−2^ s^−1^ at 25 °C, 4 °C, and 42 °C. Values are means ± SE (*n* = 5). Red asterisks indicate significant differences between 25 °C and 42 °C, and blue asterisks indicate significant differences between 25 °C and 4 °C.

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
