# Peer review of "Coordination of Cyclic Electron Flow and Water–Water Cycle Facilitates Photoprotection under Fluctuating Light and Temperature Stress in the Epiphytic Orchid *Dendrobium officinale"

_plants, 2021, doi:10.3390/plants10030606_

Round 1

Reviewer 1 Report

  • The MS title should be improved. Cooperation could be done only between living and wise organisms, not between two processes. Also, "in fluctuating light" is not a proper phrase. These issues should be considered also in the abstract.
  • The readers of Plant Journal are not specialists in measurements of plants' photosynthetic efficiency, thus authors are requested to provide a paragraph about chlorophyll fluorescence measurements in the instruction and add proper references. See the researches of Kalaji et al. during the last years. 
  • More information should be given about the light quality of the used light, especially about the spectrum as compared to daily ambient light. 
  • The discussion part should be enhanced by discussing the mechanism behind the observed changes, not only a repetition of the description of the obtained results. Authors are requested to provide correlation analysis between the measured parameters to show the differences under stress conditions and also provide a graphical scheme to show their findings clearly. 
  • Many notes were embedded in the text of the attached file. 

Author Response

The MS title should be improved. Cooperation could be done only between living and wise organisms, not between two processes. Also, "in fluctuating light" is not a proper phrase. These issues should be considered also in the abstract.

Response: Thanks a lot for these suggestions. The MS title has been changed into “Coordination of cyclic electron flow and water-water cycle facilitates photoprotection under fluctuating light at temperature stresses”. “in fluctuating light” has been changed into “under fluctuating light”.

The readers of Plant Journal are not specialists in measurements of plants' photosynthetic efficiency, thus authors are requested to provide a paragraph about chlorophyll fluorescence measurements in the instruction and add proper references. See the researches of Kalaji et al. during the last years.

Response: We have described the PSI and PSII measurements in more detail.

More information should be given about the light quality of the used light, especially about the spectrum as compared to daily ambient light.

Response: In the present study, actinic light from a 635 nm light-emitting diode (LED) equipped in Dual-PAM-100 was used as actinic light for photosynthetic measurements.

The discussion part should be enhanced by discussing the mechanism behind the observed changes, not only a repetition of the description of the obtained results. Authors are requested to provide correlation analysis between the measured parameters to show the differences under stress conditions and also provide a graphical scheme to show their findings clearly.

Response: The discussion part has been revised according to this suggestion.

Many notes were embedded in the text of the attached file.

Response: These minor suggestions have been incorporated in the revised MS.

Reviewer 2 Report

The manuscript quality is good. However, the Introduction can be more precise and focused. 

Materials and method need clarity. The authors should clearly mention the following:

  1. Whether the leaf samplee was attached to the plant or detached. If detached, whole leaf or leaf discs etc. Siz of he leaf or leaf discs etc.
  2. 2. Temperature treatment is not mentioned. Whether it is temperature shock or stress?
  3. How many minutes of low/high temperature treatment, before the data were taken? 
  4. Whether, the fluorescence analysis was done at room temperature after the temperature treatments or the analysis was done continuously under the high/low temperature?

These explanations be exhaustive, so that other researchers can follow the protocol to obtain repeatable data.

The overall quality is the paper is good. 

After these revisions, the paper can be accepted for publication.

Author Response

The manuscript quality is good. However, the Introduction can be more precise and focused.

Response: The Introduction part has been revised to be more focused.

Materials and method need clarity. The authors should clearly mention the following: Whether the leaf sample was attached to the plant or detached. If detached, whole leaf or leaf discs etc. Siz of he leaf or leaf discs etc.

Response: Attached leaves were used in this study.

Temperature treatment is not mentioned. Whether it is temperature shock or stress? How many minutes of low/high temperature treatment, before the data were taken?  Whether, the fluorescence analysis was done at room temperature after the temperature treatments or the analysis was done continuously under the high/low temperature?

Response: Before measurements, plants were incubated at 42°C, 25°C and 4°C for 15 min in the dark. Afterwards, photosynthetic measurements were conducted at each temperature.

These explanations be exhaustive, so that other researchers can follow the protocol to obtain repeatable data. The overall quality is the paper is good.  After these revisions, the paper can be accepted for publication

Response: Thanks a lot for the positive evaluation.

Reviewer 3 Report

Dear authors,

The manuscript presents an interesting problem; it is written correctly, although the authors did not avoid a few mistakes. Below are my detailed comments:

It is worth adding the research object in the title of the work.

The introduction is well written and introduces the reader to the subject. It seems to me that, despite the highly specified topic, the work can also be appreciated by a typical reader of Plants. Nevertheless, mistakes and missing words can occur, making the text difficult to understand (e.g., line 33: FROM PSII to PSI; line 38: what happens to assimilation?). I am asking the authors to carefully check the entire manuscript because such errors, although few, appear in all paragraphs.

Statistical significance must appear on the figures. It is enough if they are asterisks (blue or red) when each variant at a given point differs from the reference (25 °C). For Figure 4, the letters will look good as a marker of statistical significance. Only after adding the statistics will it be possible to determine which dependencies are significant.

Let's say I'm a typical reader and have a basic understanding of photosynthesis and chlorophyll fluorescence. How will the authors explain why the unit of the electron transport through the photosystem (ETR) is the unit of light amount on the surface over time (PPFD)?

Author Response

It is worth adding the research object in the title of the work.

Response: Thanks a lot for this important comment. We have changed the title into “Coordination of cyclic electron flow and water-water cycle facilitates photoprotection under fluctuating light at temperature stresses”.

The introduction is well written and introduces the reader to the subject. It seems to me that, despite the highly specified topic, the work can also be appreciated by a typical reader of Plants. Nevertheless, mistakes and missing words can occur, making the text difficult to understand (e.g., line 33: FROM PSII to PSI; line 38: what happens to assimilation?). I am asking the authors to carefully check the entire manuscript because such errors, although few, appear in all paragraphs.

Response: We have carefully checked the entire manuscript to avoid mistakes.

Statistical significance must appear on the figures. It is enough if they are asterisks (blue or red) when each variant at a given point differs from the reference (25 °C). For Figure 4, the letters will look good as a marker of statistical significance. Only after adding the statistics will it be possible to determine which dependencies are significant.

Response: Statistical significance has been added in Figure 4.

Let's say I'm a typical reader and have a basic understanding of photosynthesis and chlorophyll fluorescence. How will the authors explain why the unit of the electron transport through the photosystem (ETR) is the unit of light amount on the surface over time (PPFD)?

Response: The unit of PPFD is μmol photons m-2 s-1, but the unit of ETR is μmol electrons m-2 s-1.

Round 2

Reviewer 1 Report

The MS should be revised by a Native English Speaker.

There are a lot of mistakes and missed words e.g. "was not caused the change in ETRII" should be "was not caused due to the change in ETRII"

Sentences should not start with abbreviations. 

Sentences should start with capital letters.  

Author Response

The MS should be revised by a Native English Speaker.

Response: We have checked the text again and again.

There are a lot of mistakes and missed words e.g. "was not caused the change in ETRII" should be "was not caused due to the change in ETRII"

Response: After careful checking, we found some additional mistakes and revised them.

Sentences should not start with abbreviations.

Response: This mistake has been revised.

Sentences should start with capital letters.

Response: This mistake has been revised.

Reviewer 3 Report

Dear authors,

In the first comment, I noted the lack of a research object in the manuscript's title. The authors thanked for the comment, which they considered important, but did not add the research object to the title. This is called ignorance.

The authors also did not add statistics to most of the charts I asked for. The description of phenomena based on increased/decreased and more/less perhaps is valuable. Nevertheless, in my opinion, in science, the most important thing is to say whether the observed changes are significant or not. Apparently, the authors believe otherwise because they ignored my remarks again.

I believe the manuscript is valuable. I also believe that following my remarks would significantly improve its quality. However, I do not see the point in forcibly obstructing, and because authors ignore my comments, they will pass their opinion directly to the Editor.

Author Response

In the first comment, I noted the lack of a research object in the manuscript's title. The authors thanked for the comment, which they considered important, but did not add the research object to the title. This is called ignorance.

Response: We have changed the title into “Coordination of cyclic electron flow and water-water cycle facilitates photoprotection under fluctuating light at temperature stresses” in the first revision.

The authors also did not add statistics to most of the charts I asked for. The description of phenomena based on increased/decreased and more/less perhaps is valuable. Nevertheless, in my opinion, in science, the most important thing is to say whether the observed changes are significant or not. Apparently, the authors believe otherwise because they ignored my remarks again.

Response: Thanks a lot for these comments. I have added statistical analysis in Figures 2-7. Asterisks indicate significant differences between 4°C and other temperatures (25°C and 42°C). 

I believe the manuscript is valuable. I also believe that following my remarks would significantly improve its quality. However, I do not see the point in forcibly obstructing, and because authors ignore my comments, they will pass their opinion directly to the Editor.

Response: We have incorporated the Reviewer’s comments into our revised MS.